# Effect of Vitamin B12 Replacement Intervals on Clinical Symptoms and Laboratory Findings in Gastric Cancer Patients after Total Gastrectomy

**DOI:** 10.3390/cancers15204938

**Published:** 2023-10-11

**Authors:** Sin Hye Park, Sang Soo Eom, Hyewon Lee, Bang Wool Eom, Hong Man Yoon, Young-Woo Kim, Keun Won Ryu

**Affiliations:** 1Center for Gastric Cancer, National Cancer Center, Goyang 10322, Republic of Korea; 13606@ncc.re.kr (S.H.P.); i0536@paik.ac.kr (S.S.E.); kneeling79@ncc.re.kr (B.W.E.); red10000@ncc.re.kr (H.M.Y.); youngwookim@ncc.re.kr (Y.-W.K.); 2Center for Hematologic Malignancy, National Cancer Center, Goyang 10322, Republic of Korea; hwlee@ncc.re.kr

**Keywords:** gastric neoplasm, gastrectomy, vitamin B12 deficiency, anemia

## Abstract

**Simple Summary:**

The management of vitamin B12 deficiency in patients who underwent total gastrectomy remains unclear. This study evaluated the effect of vitamin B12 replacement intervals on the clinical symptoms and laboratory findings in gastric cancer patients with vitamin B12 deficiency after total gastrectomy. The vitamin B12 levels after replacement were significantly higher in the regular replacement group than in the lab-based replacement group. The post-replacement serum hemoglobin levels and symptoms were comparable between the two groups.

**Abstract:**

The management of patients with vitamin B12 deficiency after total gastrectomy (TG) remains controversial. We aimed to evaluate the effect of vitamin B12 replacement intervals on the clinical characteristics in these patients. The data from patients who received vitamin B12 supplementation after TG between 2007 and 2018 at the National Cancer Center, Korea, were retrospectively evaluated. Vitamin B12 deficiency was defined as a serum vitamin B12 level of <200 pg/mL or urine methylmalonic acid level > 3.8 mg/gCr. The patients were divided into a regular replacement group (patients received an intramuscular injection or oral medication regularly), and a lab-based replacement group (patients received vitamin B12 intermittently after checking the level). The symptoms and biochemical parameters were compared between these groups. The regular and lab-based replacement groups included 190 and 216 patients, respectively. The median vitamin B12 replacement intervals were 1 and 9 months, respectively (*p* < 0.001). After replacement, the regular replacement group had higher vitamin B12 levels than the lab-based replacement group (*p* < 0.001). However, the serum hemoglobin level showed no significant changes. After replacement, there was no significant difference in the proportion of the symptomatic patients between the groups. Replacing vitamin B12 with a lab-based protocol may be sufficient for TG patients.

## 1. Introduction

As the incidence of esophagogastric junction and cardia cancers is rising worldwide [1,2,3,4,5], the clinical practice of total gastrectomy (TG) is increasing consequently. Patients with gastric cancer who undergo TG have a high risk of developing vitamin B12 deficiency [6,7,8]. Therefore, clinical interest in vitamin B12 deficiency after TG is increasing, although its management remains controversial.

Several studies have reported that the rates of vitamin B12 deficiency were 78.0–100% in TG patients [6,9,10] and 3.2–15.7% in subtotal gastrectomy patients within 4 years post-surgery [6,8]. The causes of vitamin B12 deficiency in gastrectomy patients include the decreased secretion of hydrochloric acid and a lack of intrinsic factor [11,12,13]. In particular, intrinsic factor affects the release of proteins bound to vitamin B12, leading to the absorption of vitamin B12 [14]. The symptoms of vitamin B12 deficiency range from mild manifestations, including fatigue, dizziness, glossitis, and numbness, to severe manifestations such as gait disturbance and areflexia [15,16]. The most common neurological symptoms caused by vitamin B12 deficiency are symmetric paresthesia or numbness in the hands and feet. Severe neurological manifestations such as impaired sense of vibration, ataxia, and weakness are caused by progressive demyelination [17,18]. These symptoms can worsen the quality of life in cancer survivors.

After TG, evaluations of the vitamin B12 and methylmalonic acid (MMA) levels are recommended. In the case of vitamin B12 deficiency, supplementation is recommended by intramuscular injection of cyanocobalamin (1000 mcg) monthly or bi-monthly, or daily oral administration of mecobalamin (1000–1500 mcg) [19,20]. However, there is no consensus concerning vitamin B12 management, such as the intervals of its measurements and administration after gastrectomy [21,22,23,24]. Most patients with vitamin B12 deficiency are asymptomatic. In addition, monthly intramuscular injection of cyanocobalamin and daily oral intake of mecobalamin causes inconvenience to patients.

Therefore, we aimed to investigate the clinical symptoms, vitamin B12, and other biochemical marker levels for different replacement intervals in TG patients with vitamin B12 deficiency.

## 2. Materials and Methods

### 2.1. Study Design and Patients

Data from patients who underwent TG for gastric cancer between January 2007 and December 2018 at the National Cancer Center, Korea, were retrospectively reviewed. After TG, the patients were followed up regularly according to the follow-up schedules in our institute based on pathological stage [25], because an optimal follow-up strategy based on the pathological stage has not been established [26,27]. During the follow-up period, the complete blood count was measured every 6 months after TG. The levels of vitamin B12 and urine MMA were evaluated 1 year after surgery and every 6 months or 1 year thereafter to detect vitamin B12 deficiency.

The TG patient demographics and clinicopathologic characteristics were evaluated. Among the 1656 TG patients, 406 patients who were treated with vitamin B12 replacement therapy were included based on the postoperative vitamin B12 deficiency (*n* = 315), high urine MMA level (*n* = 51), or symptoms related to vitamin B12 deficiency (*n* = 40). The patients with the following characteristics were excluded: pre-operative vitamin B12 deficiency (*n* = 9), adjuvant or palliative chemotherapy to rule out chemotherapy-induced peripheral neuropathy (*n* = 810), loss to follow-up within 2 years post-surgery (*n* = 233), no vitamin B12 deficiency during the follow-up period (*n* = 161), vitamin B12 deficiency with iron deficiency anemia (*n* = 12), and incomplete data on vitamin B12 replacement protocol (*n* = 25) (Figure 1).

The vitamin B12-deficient patients were divided into two groups: the patients who received an intramuscular injection of 1000 mcg cyanocobalamin monthly or every 3 to 6 months and oral mecobalamin of 1500 mcg daily (regular replacement group), and those who were administered an intramuscular injection of 1000 mcg cyanocobalamin intermittently after checking the level (lab-based replacement group).

The eighth edition of the International Union for Cancer Control/American Joint Committee on Cancer staging system was used to classify the gastric cancer stage [28].

The biochemical parameters were measured during the treatment period starting from the first vitamin B12 replacement (baseline) and included the hemoglobin, serum vitamin B12, urine MMA, and mean cell volume (MCV). Vitamin B12 deficiency was defined as a serum vitamin B12 level < 200 pg/mL. A urine MMA level of 3.8 mg/gCr or higher, even in the presence of a vitamin B12 level of 200 pg/mL or higher, was considered vitamin B12 deficiency because an elevated MMA level is a more sensitive and specific parameter for the diagnosis of vitamin B12 deficiency than vitamin B12 level [29,30]. Anemia was defined using the World Health Organization criteria (hemoglobin level < 12 g/dL in women and, <13 g/dL in men) [31]. An MCV level > 100 fL was considered indicative of anemia caused by vitamin B12 deficiency. The hemoglobin and MCV were measured using Sysmex XE-2100 (Sysmex Corp. Kobe, Japan). The patients were evaluated for symptoms related to vitamin B12 deficiency in the outpatient clinic. Vitamin B12-deficient symptoms included tingling, cramping, dizziness, and glossitis [15,16].

All the study procedures were conducted in accordance with the principles of the 1964 Declaration of Helsinki. Informed consent from the patients was not required due to the retrospective study design. This study was approved by the Institutional Review Board of National Cancer Center, Korea (IRB number: NCC2022-0037).

### 2.2. Statistical Analysis

Statistical significance of the categorial variables was evaluated using Pearson’s χ^2^ test or Fisher’s exact test. The statistical differences in continuous variables between the two groups were assessed using Student’s *t*-test or the Mann–Whitney U test. The biochemical variables were analyzed using a linear mixed model for repeated measures to compare changes from baseline over time between the two groups. The least-squares means and standard errors of each variable were calculated with a first-order autoregressive covariance structure, using the visit, group, and visit-by-group interaction as fixed effects and with adjustment for the baseline value. *p* < 0.05 was considered statistically significant. All analyses were conducted using SAS software, version 9.4 (SAS Institute, Cary, NC, USA) and R version 4.2.1 (R foundation for Statistical Computing, Vienna, Austria).

## 3. Results

### 3.1. Patient Clinicopathologic Characteristics

Of 406 patients, 190 were included in the regular replacement group and 216 in the lab-based replacement group. The mean follow-up period was 62.2 ± 11.6 months (range: 24–108 months). There were no significant differences in sex, age, and pathologic stage between the two groups (Table 1). The interval from surgery to vitamin B12 replacement was shorter in the regular replacement group than in the lab-based replacement group (24.0 vs. 30.0 months, *p* = 0.032). The median follow-up interval of the laboratory tests after replacement was 12 months in both groups (*p* = 0.141). However, there was a significant difference in the median vitamin B12 replacement interval (1 month in the regular replacement group vs. 9 months in the lab-based replacement group, *p* < 0.001).

### 3.2. Laboratory Results before and after Vitamin B12 Replacement

Table 2 shows the laboratory findings at baseline and after vitamin B12 replacement. The vitamin B12 levels at baseline were not different between the regular replacement and lab-based replacement groups (103.0 vs. 117.0 pg/mL, respectively, *p* = 0.516). After vitamin B12 administration, vitamin B12 levels were increased in both groups. However, vitamin B12 levels in the regular replacement group were notably higher than those in the lab-based group during the treatment period (*p* < 0.001). Conversely, the urine MMA levels were decreased after vitamin B12 was administered. In particular, the urine MMA level in the regular replacement group was markedly decreased starting from 6 months after replacement compared with that in the lab-based replacement group (3.2 mg/gCr vs. 6.2 mg/gCr, respectively, *p* = 0.001). However, no difference was observed in the serum hemoglobin level and MCV during the treatment period between the two groups.

### 3.3. Changes in Biochemical Parameters after Vitamin B12 Replacement

The changes in each variable in the regular and lab-based replacement groups are shown in Figure 2. The regular replacement group had significantly higher vitamin B12 and lower urine MMA levels than the lab-based replacement group (*p* < 0.05) (Figure 2A,B). However, the serum hemoglobin levels and MCV were comparable between the two groups (*p* > 0.05) (Figure 2C,D).

### 3.4. Vitamin B12 Deficiency Symptoms before and after Vitamin B12 Replacement

At the beginning of the vitamin B12 administration, the patients in the regular replacement group complained of more clinical symptoms of vitamin B12 deficiency than those in the lab-based replacement group (30.0% vs. 19.4%, *p* = 0.018) (Table 3). Of the symptoms related to vitamin B12 deficiency, a tingling sensation (85/99, 85.9%) was the most common at the beginning of vitamin B12 replacement, followed by dizziness (6/99, 6.1%), cramping (6/99, 6.1%), and glossitis (2/99, 2.0%). However, no significant difference was found in the proportion of patients with symptoms between the two groups starting from 6 months after vitamin B12 replacement.

## 4. Discussion

In the current study, we evaluated the differences in the clinical symptoms and biochemical parameters related to vitamin B12 deficiency between the regular and lab-based replacement groups during the vitamin B12 replacement period after TG. The interval from TG to vitamin B12 replacement was shorter in the regular replacement group than in the lab-based replacement group. The follow-up laboratory tests after replacement were usually performed every 12 months in both groups; however, vitamin B12 was replaced more frequently in the regular replacement group than in the lab-based replacement group. However, there was no difference in the proportion of patients with clinical symptoms between the two groups during the treatment period. In the regular replacement group, vitamin B12 levels increased rapidly and reached normal levels, compared to lab-based replacement group. However, the hemoglobin and MCV levels of both groups were maintained within the normal range. To the best of our knowledge, our study is the first report to address the changing patterns of vitamin B12 and other biomarker levels following regular and lab-based replacement.

Vitamin B12 acts as a coenzyme for neurotransmitter synthesis and nerve regeneration [32], and its deficiency causes peripheral neuropathy. The mechanisms underlying chemotherapy-induced peripheral neuropathy are not fully understood. It occurs in approximately 30–80% of patients treated with antineoplastic drugs such as platinum compounds, taxanes, and vinca alkaloids [33,34]. Several studies have reported functional vitamin B12 deficiency, which manifests as a temporary decrease in vitamin B12 due to oxidative stress caused by chemotherapy, even if vitamin B12 levels are normal [35,36,37,38]. Because it is difficult to differentiate between vitamin B12 deficiency and chemotherapy-induced peripheral neuropathy in patients treated with chemotherapy, patients who received adjuvant or palliative chemotherapy were excluded from this analysis.

Patients with atrophic autoimmune gastritis have a high incidence of vitamin B12 deficiency, although the prevalence of atrophic autoimmune gastritis is 0.5–2%, and the prevalence of pernicious anemia due to this disease is 0.15–1% [39,40]. Despite the rare disease, recognizing the potential deficiency in these patients is paramount, as untreated vitamin B12 deficiency can have serious consequences for the overall health and well-being of cancer survivors.

Gastric acid, pepsin, and particularly intrinsic factor are crucial for vitamin B12 absorption, and vitamin B12 deficiency is directly associated with the extent of gastric resection [12,14]. Specifically, because of the intrinsic factor deficiency after TG, the vitamin B12 level gradually decreases after this procedure. It was reported that 78.0–100% of TG patients develop vitamin B12 deficiency within 4 years, which can occur as early as 1 year after TG [6,9,10]. In this study, the incidence of vitamin B12 deficiency within 5 years of TG was 73.5%, and the mean interval from TG to vitamin B12 deficiency identification was 30 months. There was a difference in the time interval from TG to vitamin B12 replacement between the two groups, which suggests that the regular replacement group may have started replacement early due to more symptoms of vitamin B12 deficiency than the lab-based replacement group.

It has been generally advised that the appropriate treatment for vitamin B12 deficiency is the intramuscular administration of cyanocobalamin every 1–2 months [19,20]. In addition, oral mecobalamin is an alternative treatment option for vitamin B12 deficiency [7,41,42]. However, there is no evidence or established guideline for the interval between vitamin B12 replacement and monitoring after TG. According to a nationwide survey on nutritional follow-up in Korea, most clinicians performed an anemia work-up, including a vitamin B12 level assessment, every 6 months in the 2 years after TG, and every year thereafter [24].

Although vitamin B12 levels in the lab-based replacement group were lower than those in the regular replacement group, the vitamin B12 level in the lab-based replacement group was maintained at approximately 200 pg/mL, which is the lower limit of the reference value. Furthermore, the hemoglobin levels and the proportion of patients with symptoms after replacement did not differ between the two replacement groups. These findings imply that the vitamin B12 replacement in the regular replacement group exceeded the physiologic need.

The follow-up intervals for the laboratory tests after replacement were similar in both groups. However, patients in the regular replacement group were treated more frequently than those in the lab-based replacement group. Although vitamin B12 replacement has very few side effects, it appears that there is no need for the regular replacement of vitamin B12 considering the inconveniences associated with the medical approach for vitamin B12 replacement and the cost of regular replacement. Our findings indicate that replacement should not be performed more frequently than necessary, as in the regular replacement group.

Although several studies have reported the clinical manifestations of vitamin B12 deficiency [41,43], it is difficult to distinguish between post-gastrectomy symptoms and those caused by vitamin B12 deficiency. In the present study, most of the patients were asymptomatic. Moreover, the symptomatic patients in the lab-based replacement group showed improvement after intermittent replacement (78.6%, 33/42). Even though the grading of the symptoms was not assessed, the clinical manifestations of vitamin B12 deficiency were mild and could be relieved even without frequent replacement.

Proximal gastrectomy (PG) could be a treatment method for early-stage gastric cancer located in the upper stomach. PG has several benefits over TG regarding postoperative nutrition and prevention of anemia due to the preservation of gastric volume [44,45,46]. Previous reports showed that the PG group had higher vitamin B12 levels at 1 year after surgery and a lower proportion of patients requiring vitamin B12 replacement at 2 years post-operation than the TG group [45,47,48,49,50].

Several studies demonstrated that the incidences of reflux (9.6–32%) and anastomotic stricture (15.4–35.1%) in PG with esophagogastrostomy were significantly higher than those in TG [44,51,52]. In contrast, double tract reconstruction following PG is comparable to TG in preventing reflux and anastomotic stricture [45,47,50,53,54]. However, double tract reconstruction requires complicated procedures due to the additional anastomoses. Moreover, double tract reconstruction poses difficulty in the endoscopic evaluation of the distal stomach because of the long jejunal segment between the esophagus and the remnant stomach. In particular, in countries with a high incidence of gastric cancer, including Korea and Japan, it is indispensable to evaluate the remnant stomach for metachronous remnant gastric cancer [55]. Although PG is superior to TG in terms of maintaining the vitamin B12 levels, the surgeon might hesitate to select this surgical procedure due to the pitfalls of PG.

Because the regular replacement group had different intervals of vitamin B12 supplementation, the regular replacement group was divided into two groups: the patients who received an intramuscular injection of 1000 mcg cyanocobalamin monthly or oral mecobalamin of 1500 mcg daily (routine replacement group, *n* = 110) and those who were administered an injection every 3 to 6 months (interval replacement group, *n* = 80). The laboratory findings and clinical symptoms were further analyzed among three groups (Appendix A). While the routine replacement group showed higher vitamin B12 and lower MMA levels than the interval and lab-based replacement groups, the hemoglobin and MCV levels were comparable among the three groups. The proportion of the patients with clinical symptoms at the beginning and 6 months after replacement of vitamin B12 was higher in the routine replacement group than that of the other two groups; however, the proportion of symptomatic patients was similar thereafter. In other words, it is sufficient to replace vitamin B12 every 3–6 months or through a lab-based protocol in vitamin B12-deficient patients after TG. Based on our results, we recommend checking the vitamin B12 levels and replacing it at a visit every 6 months for TG patients, rather than every month.

There were several limitations in this study. First, this was a retrospective analysis conducted in a single institution with a relatively small number of patients, which may have contributed to the selection bias. Second, the regular replacement group was heterogenous, including patients who were replaced with injected cyanocobalamin at different monthly intervals. Third, the data on pre-operative vitamin B12 levels were not included. Hence, we could not analyze the changes in the biochemical parameters before and after TG. In addition to the changes in the hemoglobin levels, the cumulative incidence of anemia after vitamin B12 replacement was not analyzed. Finally, iron deficiency anemia, the most common cause of anemia after gastrectomy, was not evaluated. Some patients may have vitamin B12 deficiency combined with iron deficiency anemia, but the association between the two has not been fully investigated.

## 5. Conclusions

In the present study, a significant difference was observed in the level of vitamin B12 between different vitamin B12 replacement intervals after TG, while anemia and symptoms of vitamin B12 deficiency were comparable between the two groups. However, patients should visit clinics for replacements more frequently. Based on our results, it might be sufficient to replace vitamin B12 with a lab-based protocol in TG patients rather than using a regular replacement protocol. Nevertheless, further studies are needed to determine the appropriate treatment and management strategies for vitamin B12 deficiency in TG patients.

## Figures and Tables

**Figure 1 cancers-15-04938-f001:**
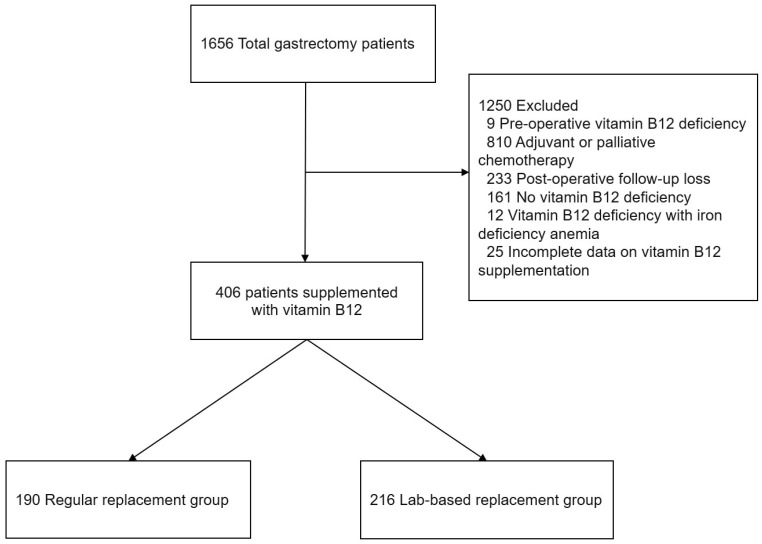
Flowchart of the study.

**Figure 2 cancers-15-04938-f002:**
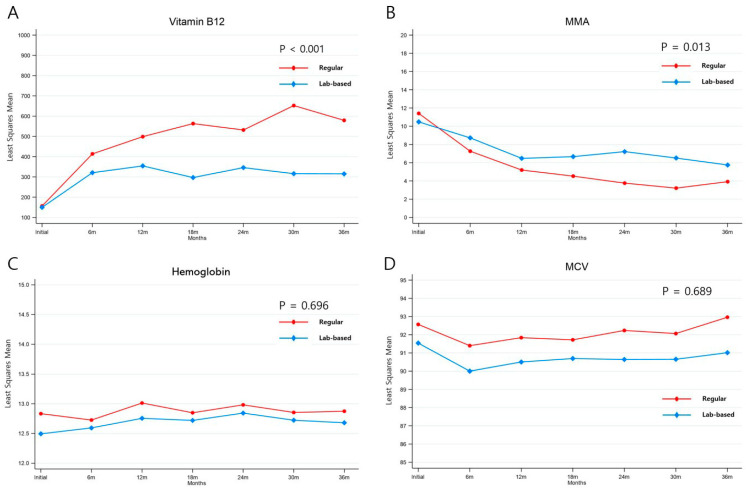
Changes in biochemical parameters after vitamin B12 replacement. (**A**) Vitamin B12, (**B**) MMA, (**C**) hemoglobin, (**D**) MCV. Abbreviations: MMA, methylmalonic acid; MCV, mean cell volume.

**Table 1 cancers-15-04938-t001:** Patient clinicopathologic characteristics.

Variables	Regular Replacement Group (*n* = 190)	Lab-based Replacement Group (*n* = 216)	*p*-Value
Sex			0.921
Male	139 (73.2)	156 (72.2)	
Female	51 (26.8)	60 (27.8)	
Age * (years)	58.5 (50.0–68.0)	59.0 (48.5–66.0)	0.271 *
Pathologic stage			0.340
I	163 (85.8)	176 (81.5)	
II	20 (10.5)	32 (14.8)	
III	7 (3.7)	6 (2.8)	
IV	0 (0.0)	2 (0.9)	
Time interval from surgery to vitamin B12 replacement * (month)	24.0 (21.0–36.0)	30.0 (24.0–36.0)	0.032 *
Follow-up interval for laboratory tests after replacement * (month)	12.0 (6.0–12.0)	12.0 (6.0–12.0)	0.141 *
Vitamin B12 replacement interval * (month)	1.0 (1.0–3.0)	9.0 (6.0–12.0)	<0.001 *

Values are *n* (%) unless otherwise indicated. * Values are presented as medians (25th and 75th percentiles).

**Table 2 cancers-15-04938-t002:** Laboratory results before and after vitamin B12 replacement.

Variables	Regular Replacement Group (*n* = 190)	Lab-Based Replacement Group (*n* = 216)	*p*-Value
Vitamin B12			
Baseline ^†^	103.0 (99.0–184.5)	117.0 (99.0–170.0)	0.516
6 months after replacement	340.0 (197.5–577.5)	175.0 (99.0–358.0)	<0.001
12 months after replacement	351.0 (207.0–687.0)	176.0 (99.0–444.0)	<0.001
18 months after replacement	488.0 (274.0–809.5)	213.0 (99.0–340.0)	<0.001
24 months after replacement	389.0 (254.5–752.5)	226.5 (107.0–436.0)	<0.001
30 months after replacement	515.5 (205.0–987.0)	175.0 (99.0–266.5)	<0.001
36 months after replacement	407.0 (223.5–789.0)	183.0 (115.0–357.0)	<0.001
Urine MMA			
Baseline ^†^	8.2 (4.6–14.6)	7.6 (4.8–13.2)	0.637
6 months after replacement	3.2 (2.0–6.3)	6.2 (2.7–10.5)	0.001
12 months after replacement	3.0 (1.8–6.5)	4.6 (1.9–9.3)	0.094
18 months after replacement	2.6 (1.9–4.0)	5.2 (2.8–9.2)	<0.001
24 months after replacement	3.3 (2.0–4.9)	4.5 (1.7–10.0)	0.053
30 months after replacement	2.2 (1.6–3.3)	5.6 (2.7–7.7)	<0.001
36 months after replacement	2.3 (1.5–3.2)	4.8 (2.1–7.6)	0.001
Hemoglobin			
Baseline ^†^	13.0 (11.7–13.8)	12.6 (11.6–13.6)	0.06
6 months after replacement *	12.7 (1.4)	12.6 (1.5)	0.27
12 months after replacement	13.2 (12.1–13.9)	12.8 (11.9–13.8)	0.092
18 months after replacement *	12.8 (1.2)	12.7 (1.3)	0.306
24 months after replacement *	13.0 (1.3)	12.8 (1.4)	0.291
30 months after replacement	12.9 (11.9–14.0)	12.6 (11.5–13.6)	0.123
36 months after replacement *	12.8 (1.5)	12.6 (1.5)	0.328
MCV			
Baseline ^†^	92.8 (88.9–96.1)	91.8 (87.5–96.0)	0.183
6 months after replacement	91.2 (88.3–94.8)	91.2 (86.8–95.0)	0.25
12 months after replacement	91.7 (88.8–95.4)	91.5 (87.4–94.3)	0.122
18 months after replacement	91.8 (88.6–95.7)	91.3 (87.8–95.0)	0.276
24 months after replacement	92.0 (88.9–95.9)	91.1 (87.1–94.5)	0.05
30 months after replacement	92.2 (88.6–95.2)	91.2 (87.2–94.8)	0.218
36 months after replacement	92.4 (88.8–95.6)	90.9 (87.5–94.5)	0.054

Values are medians (25th and 75th percentiles) unless otherwise indicated. * Values are means (standard deviations). ^†^ First vitamin B12 replacement. Abbreviations: MMA, methylmalonic acid; MCV, mean cell volume.

**Table 3 cancers-15-04938-t003:** Vitamin B12 deficiency symptoms before and after vitamin B12 replacement.

Variable	Regular Replacement Group (*n* = 190)	Lab-Based Replacement Group (*n* = 216)	*p*-Value
Symptoms at pre-replacement	*n* = 190	*n* = 216	0.018
Absence	133 (70.0)	174 (80.6)	
Presence	57 (30.0)	42 (19.4)	
Symptoms 6 months after replacement	*n* = 182	*n* = 211	0.282
Absence	155 (85.2)	188 (89.1)	
Presence	27 (14.8)	23 (10.9)	
Symptoms 12 months after replacement	*n* = 173	*n* = 204	0.283
Absence	153 (88.4)	186 (91.2)	
Presence	20 (11.6)	18 (8.8)	
Symptoms 18 months after replacement	*n* = 155	*n* = 175	0.746
Absence	133 (85.8)	155 (88.6)	
Presence	22 (14.2)	20 (11.4)	
Symptoms 24 months after replacement	*n* = 138	*n* = 158	0.871
Absence	124 (89.9)	139 (88.0)	
Presence	14 (10.1)	19 (12.0)	
Symptoms 30 months after replacement	*n* = 107	*n* = 108	0.307
Absence	92 (86.0)	97 (89.8)	
Presence	15 (14.0)	11 (10.2)	
Symptoms 36 months after replacement	*n* = 95	*n* = 100	0.71
Absence	85 (89.5)	91 (91.0)	
Presence	10 (10.5)	9 (9.0)	

Values are *n* (%).

## Data Availability

The data presented in this study are available on request from the corresponding author.

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
