# Peer review of "Effect of Vitamin B12 Replacement Intervals on Clinical Symptoms and Laboratory Findings in Gastric Cancer Patients after Total Gastrectomy"

_cancers, 2023, doi:10.3390/cancers15204938_

Round 1
Reviewer 1 Report
Congratulations to the authors, excellent article, I have no particular observations except the obvious non-generabilizability of the data considering the retrospective design.
Author Response
Thank you for comments.
Reviewer 2 Report
The manuscript “Effect of Vitamin B12 Replacement Intervals on Clinical Symptoms and Laboratory Findings in Gastric Cancer Patients after Total Gastrectomy “ appears to be a well-conducted study addressing an important clinical question. It is solid in terms of methodology and data presentation. However, it may not be considered particularly innovative, and there is room for further improvement in the discussion of clinical significance. However, while the study does not appear to introduce a particularly innovative as it simply primarily assesses the impact of different replacement intervals on vitamin B12 levels and symptoms, which is an important but relatively conventional research question, it could be improved by introducing and highlighting the historical misconception regarding vitamin B12 supplementation in neoplastic patients and the importance of evaluating potential deficiencies, particularly in those with autoimmune gastritis. Indeed, In the past, there has been a historical misconception among clinical physicians regarding the administration of vitamin B12 to neoplastic patients, stemming from a belief that vitamin B12 could stimulate cell growth. However, it is crucial to reassess this perspective in light of evolving medical knowledge. On the contrary, for patients with gastric malignancies, especially those with coexisting autoimmune atrophic gastritis (who are already at an increased risk), the likelihood of vitamin B12 deficiency is even higher (see also PMID: 28216963, PMID: 30236765, PMID: 34454714 ). Recognizing the potential deficiency in these patients is of paramount importance, as untreated vitamin B12 deficiency can have serious consequences for the overall health and well-being of cancer survivors. Hence, it is imperative to consider regular monitoring of vitamin B12 levels in neoplastic patients, especially those with underlying autoimmune gastritis, to ensure timely intervention and prevent complications associated with vitamin B12 deficiency. This addition helps provide context for the historical perspective on vitamin B12 supplementation in neoplastic patients and underscores the importance of reevaluating this approach in light of current understanding, particularly for patients at higher risk due to autoimmune gastritis.
Minor comments:
- - While the manuscript reports statistical significance (e.g., p-values), it may be helpful to include effect sizes or clinical significance when discussing the practical implications of the findings. This can provide readers with a better sense of the magnitude of the observed differences.
- - The overall quality of writing is sufficient
The overall quality of writing is sufficient
Author Response
Major comments:
Response) Thank you for your kind comments. As you commented, patients with atrophic autoimmune gastritis have a high incidence of vitamin B12 deficiency, which can affect their long-term quality of life.
However, the prevalence of atrophic autoimmune gastritis is 0.5%-2%, and the prevalence of pernicious anemia due to this disease is 0.15%-1%, which is very rare.
Considering that the patients included in the current study did not have autoimmune gastritis and that atrophic autoimmune gastritis is very rare in Asian countries, it is not appropriate to address vitamin B12 replacement in patients with atrophic autoimmune gastritis in the current study including the general cancer population.
Minor comments:
Response) Thank you for your comments. We have revised sentences as you mentioned.
Revised)
Discussion (8pages)
In the current study, we evaluated the differences in the clinical symptoms and biochemical parameters related to vitamin B12 deficiency between the regular and lab-based replacement groups during the vitamin B12 replacement period after TG. The interval from TG to vitamin B12 replacement was shorter in the regular replacement group than in the lab-based replacement group. Follow-up laboratory tests after replacement were usually performed every 12 months in both groups; however, vitamin B12 was replaced more frequently in the regular replacement group than in the lab-based replacement group. However, there was no difference in the proportion of patients with clinical symptoms between the two groups during the treatment period. In the regular replacement group, vitamin B12 levels increased rapidly and reached normal levels, compared to lab-based replacement group. However, the hemoglobin and MCV levels of both groups were maintained within the normal range. To the best of our knowledge, our study is the first report to address the changing patterns of vitamin B12 and other biomarker levels after replacement after regular and lab-based replacement.
Reviewer 3 Report
Authors evaluated the effect of vitamin B12 replacement intervals between two groups, and concluded that replacing vitamin B12 with a lab-based protocol may be sufficient for TG patients.
The aim of the study had clinical usage, however there were some questions as below.
The 'regular replacement group' was replaced Vit. B12 by three kinds of method, such as intramuscular injection monthly or every 3-6 months and oral intake daily . On the other hand the 'lab-based replacement group' was replaced only by intramucular injection after checking the level of Vit.B12. (page.3,line.87-91). Authors were recommend to examine and discuss more in detail about which interval method was suitable among four kinds of interval methods.
Author Response
Response) Thank you for your kind comments. According to a nationwide survey conducted on surgeons in Korea [1], it was reported that anemia work-up is usually performed every 6 months within 3 years, and every 12 months thereafter. At our institution, EGC patients are followed up every 6 months after surgery, and AGCs are followed up every 3 months during the first 3 years and every 6 months for the next 2 years [2]. Based on our study results and the above, we recommend checking the vitamin B12 levels and replacing it at visit every 6 months for total gastrectomy patients, rather than every month. We have added this recommendation in discussion part.
References
- Kim T-H, Kim I-H, Kang SJ, Choi M, Kim B-H, Eom BW, et al. Korean Practice Guidelines for Gastric Cancer 2022: An Evidence-based, Multidisciplinary Approach. J Gastric Cancer. 2023;23:3–106.
- Eom BW, Ryu KW, Lee JH, Choi IJ, Kook MC, Cho SJ, et al. Oncologic effectiveness of regular follow-up to detect recurrence after curative resection of gastric cancer. Ann Surg Oncol. 2011;18:358–364.
Revised)
Discussion (9pages)
Several studies demonstrated that the incidences of reflux (9.6–32%) and anastomotic stricture (15.4–35.1%) in PG with esophagogastrostomy were significantly higher than those in TG [42, 49, 50]. In contrast, double tract reconstruction following PG is comparable to TG in preventing reflux and anastomotic stricture [43, 45, 48, 51, 52]. However, double tract reconstruction requires complicated procedures due to additional anastomoses. Moreover, double tract reconstruction poses difficulty in endoscopic evaluation of the distal stomach because of the long jejunal segment between the esophagus and the remnant stomach. In particular, in countries with a high incidence of gastric cancer, including Korea and Japan, it is indispensable to evaluate the remnant stomach for metachronous remnant gastric cancer [53]. Although PG is superior to TG in terms of maintaining vitamin B12 levels, the surgeon might hesitate to select this surgical procedure due to the pitfalls of PG.
Based on our study results and the above, we recommend checking the vitamin B12 levels and replacing it at visit every 6 months for total gastrectomy patients, rather than every month.
There were several limitations in this study. First, this was a retrospective analysis conducted in a single institution with a relatively small number of patients, which may have contributed to the selection bias. Second, the regular replacement group was het-erogenous, including patients who were replaced with injected cyanocobalamin at different monthly intervals. Third, the data on pre-operative vitamin B12 levels were not included. Hence, we could not analyze the changes in biochemical parameters before and after TG. In addition to changes in hemoglobin levels, the cumulative incidence of anemia after vitamin B12 replacement was not analyzed. Finally, iron deficiency anemia, the most common cause of anemia after gastrectomy, was not evaluated. Some patients may have vitamin B12 deficiency combined with iron deficiency anemia, but the association between the two has not been fully investigated.
Reviewer 4 Report
Dear Authors, congratulations for the research, it is an excellent work, high quality scientific, high evidence and useful for current practice
Author Response
Thank you for comments.
Round 2
Reviewer 2 Report
The authors did not fully address my feedback regarding the inclusion of the historical context and the importance of reassessing the administration of vitamin B12 to neoplastic patients, especially those with gastric malignancies. Even if their study primarily focuses on patients without autoimmune gastritis or if it is rare in Asia, discussing these aspects in the discussion section can make the work more universally applicable and valuable to a broader international audience. Therefore I reiterate my suggestion to include in the discussion these considerations:
“….for patients with gastric malignancies, especially those with coexisting autoimmune atrophic gastritis (who are already at an increased risk), the likelihood of vitamin B12 deficiency is even higher (see also PMID: 28216963, PMID: 30236765, PMID: 34454714 ). Recognizing the potential deficiency in these patients is of paramount importance, as untreated vitamin B12 deficiency can have serious consequences for the overall health and well-being of cancer survivors.”
sufficient
Author Response
Response) Thank you for your kind comments. As you mentioned, we have added your suggestions in the discussion part.
Revised)
Discussion (page 8)
Vitamin B12 acts as a coenzyme for neurotransmitter synthesis and nerve regen-eration [32], and its deficiency causes peripheral neuropathy. The mechanisms under-lying chemotherapy-induced peripheral neuropathy are not fully understood. It occurs in approximately 30–80% of patients treated with antineoplastic drugs such as platinum compounds, taxanes and vinca alkaloids [33, 34]. Several studies have reported func-tional vitamin B12 deficiency, which manifests as a temporary decrease in vitamin B12 due to oxidative stress caused by chemotherapy, even if vitamin B12 levels are normal [35-38]. Because it is difficult to differentiate between vitamin B12 deficiency and chemotherapy-induced peripheral neuropathy in patients treated with chemotherapy, patients who received adjuvant or palliative chemotherapy were excluded from this analysis.
Patients with atrophic autoimmune gastritis have a high incidence of vitamin B12 deficiency, although the prevalence of atrophic autoimmune gastritis is 0.5%-2%, and the prevalence of pernicious anemia due to this disease is 0.15%-1% [39,40]. Despite the rare disease, recognizing the potential deficiency in these patients is paramount, as untreated vitamin B12 deficiency can have serious consequences for the overall health and well-being of cancer survivors.
Reference
- Cavalcoli F, Zilli A, Conte D, Massironi S. Micronutrient deficiencies in patients with chronic atrophic autoimmune gastritis: A review. World J Gastroenterol. 2017;23:563-572.
- Shah SC, Piazuelo MB, Kuipers EJ, Li D. AGA Clinical Practice Update on the Diagnosis and Management of Atrophic Gastritis: Expert Review. Gastroenterology. 2021;161:1325-1332.e1327.
Reviewer 3 Report
Authors didn' t revise according to reviewer's comments yet.
Authors were recommended to investigate about three groups below, and statistically analyze between the groups .
1. 'regular replacement group' injected intramuscullary monthly
2 . 'regular replacement group' injected intramuscullary every 3-6 months and oral intake daily
3. 'lab-based replacement group'
The 'regular replacement group' was replaced Vit. B12 by three kinds of method, such as intramuscular injection monthly or every 3-6 months and oral intake daily . On the other hand the 'lab-based replacement group' was replaced only by intramucular injection after checking the level of Vit.B12. (page.3,line.87-91). Authors were recommend to examine and discuss more in detail about which interval method was suitable among four kinds of interval methods.
Author Response
Response) Thank you for your kind comments. As you commented, we have added analysis of three subgroups in the discussion part.
Revised)
Discussion (9pages)
Several studies demonstrated that the incidences of reflux (9.6–32%) and anastomotic stricture (15.4–35.1%) in PG with esophagogastrostomy were significantly higher than those in TG [42, 49, 50]. In contrast, double tract reconstruction following PG is comparable to TG in preventing reflux and anastomotic stricture [43, 45, 48, 51, 52]. However, double tract reconstruction requires complicated procedures due to additional anastomoses. Moreover, double tract reconstruction poses difficulty in endoscopic evaluation of the distal stomach because of the long jejunal segment between the esophagus and the remnant stomach. In particular, in countries with a high incidence of gastric cancer, including Korea and Japan, it is indispensable to evaluate the remnant stomach for metachronous remnant gastric cancer [53]. Although PG is superior to TG in terms of maintaining vitamin B12 levels, the surgeon might hesitate to select this surgical procedure due to the pitfalls of PG.
Because the regular replacement group had different intervals of vitamin B12 supplementation, the regular replacement group divided into two groups: patients who received intramuscular injection of 1000 mcg cyanocobalamin monthly or oral mecobalamin of 1500 mcg daily (routine replacement group, n=110) and those who were administered injection every 3 to 6 months (interval replacement group, n=80). Laboratory findings ​​and clinical symptoms were further analyzed among three groups (Supplementary Figure 1, Supplementary Table 1). While the routine replacement group showed higher vitamin B12 and lower MMA levels than the interval and lab-based replacement groups, hemoglobin and MCV levels were comparable among the three groups. The proportion of patients with clinical symptoms at the beginning and 6 months after replacement of vitamin B12 was higher in the routine replacement group than that of the other two groups, however the proportion of symptomatic patients was similar thereafter. In other words, it is sufficient to replace vitamin B12 every 3-6 months or through a lab-based protocol in vitamin B12-deficient patients after TG. Based on our results, we recommend checking the vitamin B12 levels and replacing it at visit every 6 months for TG patients, rather than every month.
Supplementary Table 1. Vitamin B12 deficiency symptoms before and after vitamin B12 replacement between three groups.
|
Variable |
Routine replacement group (n = 110) |
Interval replacement group (n = 80) |
Lab-based replacement group (n = 216) |
p-value |
|
Symptoms at pre-replacement |
n = 110 |
n = 80 |
n = 216 |
<0.001 |
|
Absence |
65 (59.1%) |
68 (85.0%) |
174 (80.6%) |
|
|
Presence |
45 (40.9%) |
12 (15.0%) |
42 (19.4%) |
|
|
Symptoms 6 months after replacement |
n = 105 |
n = 77 |
n = 211 |
0.025 |
|
Absence |
83 (75.5%) |
72 (90.0%) |
188 (87.0%) |
|
|
Presence |
22 (20.0%) |
5 (6.2%) |
23 (10.6%) |
|
|
Symptoms 12 months after replacement |
n = 98 |
n = 75 |
n = 204 |
0.115 |
|
Absence |
83 (75.5%) |
70 (87.5%) |
186 (86.1%) |
|
|
Presence |
15 (13.6%) |
5 (6.2%) |
18 (8.3%) |
|
|
Symptoms 18 months after replacement |
n = 86 |
n = 69 |
n = 175 |
0.206 |
|
Absence |
70 (63.6%) |
63 (78.8%) |
155 (71.8%) |
|
|
Presence |
16 (14.5%) |
6 (7.5%) |
20 (9.3%) |
|
|
Symptoms 24 months after replacement |
n = 77 |
n = 61 |
n = 158 |
0.616 |
|
Absence |
67 (60.9%) |
57 (71.2%) |
139 (64.4%) |
|
|
Presence |
10 (9.1%) |
4 (5.0%) |
19 (8.8%) |
|
|
Symptoms 30 months after replacement |
n = 61 |
n = 46 |
n = 108 |
0.521 |
|
Absence |
51 (46.4%) |
41 (51.2%) |
97 (44.9%) |
|
|
Presence |
10 (9.1%) |
5 (6.2%) |
11 (5.1%) |
|
|
Symptoms 36months after replacement |
n = 53 |
n = 42 |
n = 100 |
0.73 |
|
Absence |
46 (41.8%) |
39 (48.8%) |
91 (42.1%) |
|
|
Presence |
7 (6.4%) |
3 (3.8%) |
9 (4.2%) |
|
Values are n (%)
Supplementary Figure 1. Changes in biochemical parameters after vitamin B12 replacement between three groups. (a) Vitamin B12, (b) MMA, (c) Hemoglobin, (d) MCV. Abbreviations: MMA, methylmalonic acid; MCV, mean cell volume.

Round 3
Reviewer 2 Report
This manuscript has undergone substantial revisions and improvements since its initial submission
Overall quality of the language is good